# Influence of Green Tea Added to Cherry Wine on Phenolic Content, Antioxidant Activity and Alpha-Glucosidase Inhibition during an In Vitro Gastrointestinal Digestion

**DOI:** 10.3390/foods11203298

**Published:** 2022-10-21

**Authors:** Małgorzata Lasik-Kurdyś, Małgorzata Gumienna, Barbara Górna, Noranizan Mohd Adzahan

**Affiliations:** 1Department of Food Technology of Plant Origin, Faculty of Food Science and Nutrition, Poznan University of Life Sciences, Wojska Polskiego 31, 60-624 Poznan, Poland; 2Department of Food Technology, Faculty of Food Science and Technology, Universiti Putra Malaysia, Serdang 43400, Malaysia

**Keywords:** cherry wine, green tea, polyphenols, alpha-glucosidase inhibition, in vitro gastrointestinal digestion

## Abstract

Cherries are a good source of bioactive compounds, with high antioxidant activity as well as nutritional and therapeutic importance. In this study, cherry wines enriched with green tea infusion (mild and concentrated) were produced, and their biological properties were evaluated. During winemaking, the main vinification parameters (alcohol, reducing sugars, acidity, total polyphenol content) as well biological activity (antioxidant activity, alpha-glucosidase inhibition potential) were determined. An in vitro digestion process was also performed to evaluate the impact of the gastrointestinal environment on the biological stability of the wines, and to analyze the interactions of wine-intestinal microflora. The addition of green tea to the cherry wine significantly increased the total polyphenol content (up to 2.73 g GAE/L) and antioxidant activity (up to 22.07 mM TE/L), compared with the control wine. However, after in vitro digestion, a reduction in total polyphenols (53–64%) and antioxidant activity (38–45%) were noted. Wines fortified with green tea expressed a stronger inhibition effect on intestinal microflora growth, of which *E. coli* were the most sensitive microorganisms. The tea-derived bioactive compounds significantly increased the potential of alpha-glucosidase inhibition. The proposed wines could be a good alternative type of wine, with an increased polyphenol content and the potential to control the insulin response supporting therapy for diabetes.

## 1. Introduction

Biologically active food compounds are known to be impactful on health, improving physiological functions and reducing the risk of developing lifestyle-related diseases such as atherosclerosis, diabetes, cataracts, Parkinson’s disease and Alzheimer’s disease. The largest known group of bioactive compounds found in food are polyphenolic compounds. They are found naturally in fruits, vegetables, tea and wine. Besides the nutritional benefits, polyphenols are also widely used in the food industry in the form of extracts or dried products, functioning as natural antioxidants and stabilizing compounds or as a color improver [1,2,3,4]. The specific chemical structure consisting of a multitude of hydroxyl groups provides the antioxidant properties, protecting organisms against the harmful effects of free radicals. Polyphenols also present anti-inflammatory, antibacterial, antifungal and antiviral properties. In addition, they seal blood vessels, improve the lipid profile, and delay atherosclerotic changes, clinically expressed by reducing the risk of cardiovascular diseases [2,3,5,6]. Apart from fresh fruit and vegetables, wine and tea are some of the most common food products characterized by an increased concentration of polyphenolic compounds, and their health-promoting properties have been scientifically and clinically proven for many years [7,8,9,10,11]. 

Cherry wines are attractive, due to their intense red color, rich taste and aroma. The chemical composition of cherry fruits includes: 80–90% of water, 17% of dry matter, 6–11% of sugars, 0.6–2.45% of organic acids, 0.9–1.7% of nitrogen compounds, 0.35–0.62% of minerals and 0.11–0.30% of tannins [12,13,14,15]. Cherries are also rich in polyphenolic compounds, especially anthocyanins, which are responsible for the color of cherries and have a strong antioxidant effect. During maceration, 60–80% of these compounds pass into the juice. Another group of polyphenols is flavonoids such as quercetin and campferol and their glycosidic forms. A group of compounds with similar properties are phenolic acids, and the most common of them are caffeic, cholorogenic, protocatechic and p-coumaric acid. In addition to phenolic acids, cherry wines also contain organic acids such as tartaric, malic, lactic, succinic, oxalic, acetic and citric acids. They play a key role in the quality of wines, balancing the taste, and influencing the chemical stability and acidity. The level of anthocyanins, phenolic acids and antioxidant capacity in the case of wines produced from cherries is comparable to the values of these parameters for grape wines [13,14]. The content of health-promoting compounds in cherry wine, and its characteristics such as color, taste or tartness are most influenced by the raw materials—their variety, degree of maturity, climatic conditions, soil and fertilization [7,13,16,17,18,19]. 

The health benefits of green tea have been known, mainly from Chinese medicine, for over several thousand years. Within the group of polyphenols contained in green tea, catechins and flavonoids, strongly neutralizing free radicals, deserve special attention. Bioactive ingredients contained in green tea infusions inhibit the growth and metabolic processes of cancer cells, lead to deformation of the structure of already formed cells and contribute to the occurrence of apoptosis [8,10,11]. 

In general, products containing an increased content of polyphenols have many health-promoting properties. In addition to those described above, the literature and research also indicate the effects of polyphenols in the prevention of diabetes. Polyphenolic compounds show the ability to inhibit intestinal glucosidases, i.e., enzymes of the gastrointestinal tract, digesting complex carbohydrates supplied with food: di-, oligo- and polysaccharides, e.g., starch, and preparing them for absorption in the small intestine [20]. This happens by breaking down glycosidic bonds, which results in obtaining simple sugar molecules. Inhibition of the reaction of decomposition of polysaccharides to simple sugars results in the limitation of their metabolism and reduction of their absorption. As a result, it allows the regulation of the postprandial blood glucose level. This is especially important for people with diabetes. Dietary ingredients with such specific biological activity can support patients by reducing the risk of diabetes complications in a completely natural way, without the side effects often associated with pharmacological treatment [20,21,22]. 

The aim of the study was to assess the influence of the addition of green tea to cherry wine on the biological properties of this wine. The wines were evaluated for their total polyphenol concentration, antioxidant activity, and ability to inhibit alpha-glucosidase (AGL). Subsequently, the wines were subjected to an in vitro digestion process. This allowed for the analysis of changes in the biological activity of these products during subsequences stages of gastrointestinal digestion and of their impact on the intestinal microflora.

## 2. Materials and Methods

### 2.1. Cherries

Sour cherry varieties Łutówka (*Prunus cerasus*) from the Agricultural Experimental Station of Poznan University of Life Sciences “Przybroda” (Przybroda, Poland) were picked at technological maturity and were used as the fruit material for wine production. Green tea Ceylon Dimbula OPA 2 in the form of dry leaves was also used in the performed experiment. The tea (with the country of origin Sri Lanka) was purchased in the specialist tea shop “Tea time”, in Poznań, Poland.

### 2.2. Yeast

The lyophilised wine yeasts culture *Saccharomyces cerevisiae* (Enovini WS, Browin, Łódź, Poland) was used for wine making. The applied dosage was 0.25 g/L. The yeast were first dehydrated in 50 mL of water at 30 °C for 15 min. A supplementation of yeast nutrient Brovin 401,010 containing sources of N, P and wit. B1 (thiamine) (Browin, Łódź, Poland) was also applied (dosage 0.25 g/L). According to the producer’s recommendation, vitamin B1 stimulates yeast growth and is a fermentation activator. The compounds contained in the nutrition media are used entirely by yeasts during their multiplication and fermentation, so they do not affect the taste and aroma of the wine.

### 2.3. Experimental Design

#### 2.3.1. Vinification Process

Cherries (after manual destemming), water, sugar, yeast and nutrient were placed and mixed in plastic fermentation buckets of the total volume 30 L (Biovin, Łódź, Poland). To prepare the fermentation environment for yeast as well as to avoid any contamination risks, a sulphitation process was applied (in a dosage 50 mg K_2_S_2_O_5_/L) (Sigma-Aldrich, St. Louis, MO, USA). Three different variants of vinification process were performed:

Wine A: cherry wine with green tea infusion.

Wine B: cherry wine with concentrated green tea infusion.

Wine C: cherry wine (control).

The composition of each variant is presented in Table 1.

The vinification process was conducted in plastic fermentation buckets of a total volume 30L (Biovin, Łódź, Poland) at 23 ± 1 °C. After 7 days, pressing in a wood basket press (Winiarz.pl, Zielona Góra, Poland) was conducted, to separate pomace from the young wine. This was followed by a second serving of sucrose (1.4 kg to each wine sample), as stated in Table 1. After pressing, each wine bucket was divided into 5 L fermentation glass bottles (Biovin, Poland), in which the vinification process was continued. After two months, racking of the wine from the sediment was carried out. Next, the wine bottles were placed in a cold room with a temperature of 12 ± 2 °C, and were stored in the dark for 6 months. All wine variants were made in triplicate. After 6 months of the winemaking process, the wine samples were stored at 7 °C and were directly subjected to appropriate analyses (without freezing).

#### 2.3.2. In Vitro Gastrointestinal Digestion Process

The in vitro digestion process was performed in a 500 mL glass bioreactor (New Brunswick), simulating the conditions of the human gastrointestinal tract (stomach, small and large intestine), and maintained at 37 °C. The parameters of in vitro digestion were selected according to studies carried out by Aura et al. 1999 [23], Gil-Izquierdo et al. 2001 [24] and our own modifications [25]. In the experiment, human intestinal microflora was applied. It was isolated from the feces of three healthy volunteers aged 25–30 years, and standardized according Knarreborg et al. 2002 [26]. The applied intestinal microflora included four strains: *Lactobacillus plantarum*, *Escherichia coli*, *Enterococcus faecium* and *Bifidobacterium bifidum*. The mixed culture was incubated in brain heart broth medium (BHI, Fluka, Buchs, Switzerland) for 24 h at 37 °C. Subsequently, 1 mL of the culture was centrifuged at 4125 *g* for 10 min. The obtained sediment was used in the model of the gastrointestinal tract for digested wine inoculation. The concentration of the applied inoculum was approximately 10^6^ cfu/mL. 

In the digestion experiment, 230 mL of the tested wine was applied. The diagram of the in vitro digestion process is presented in Figure 1. First, at the stomach stage, the pH of the wine sample was reduced to 2.0 by 1 M HCl (POCh, Gliwice, Poland) and then pepsin (60,000U; Sigma-Aldrich, Munich, Germany) was added (0.092 g of pepsin in 2 mL 0.1 M HCl). After 2 h incubation in pH 2.0 with pepsin, sample No. 2 was taken. The next stage of digestion was the small intestine, where the pH value was increased by using 1 M NaHCO_3_ (POCh, Gliwice, Poland). After reaching pH 6.0, the pancreatic bile extract was added. It was prepared as follows: 0.02 g of pancreatin (Sigma-Aldrich, Munich, Germany) and 0.12 g of bile salt (Sigma-Aldrich, Munich, Germany) were dissolved in 5 mL 0.1 M NaHCO_3_, each separately, and then both solutions were mixed together directly before introduction into the bioreactor. When the pH was increased to 7.4, sample No. 3 was taken. Next, intestinal microflora was implemented (prepared as described above). After 15 min of mixing, sample No. 4 was taken. Afterwards, 2 h after bacteria inoculation, sample No. 5 was taken, and the small intestine stage was finished. In the consecutive stage of the large intestine, the value of pH was increased to 8.0 by 1M NaHCO_3_ and then sample No. 6 was taken. After another 18 h, sample No. 7 was collected, as the final one. The amount of reagents (NaHCO_3_ and HCl) used for pH regulation was registered and included in the calculations. 

#### 2.3.3. Determination of the Number of Bacteria

Selective media were used for the determination of intestinal bacteria: MacConkey selective medium (Sigma-Aldrich, Munich, Germany) for *Entrobacteriaceae*, MRS medium (Sigma-Aldrich, Munich, Germany) for *Lactobacillus*, agar with kanamycin, esculin and sodium azide (Sigma-Aldrich, Munich, Germany) for *Enteroccocus* and Garch’s medium (Sigma-Aldrich, Munich, Germany) for *Bifidobacterium*. The number of cfu/mL was assessed by the direct plating method. For *Escherichia coli* and *Enterococcus faecium*, aerobic conditions, and for *Lactobacillus plantarum* a double flood method, were applied, while for *Bifidobacterium bifidum* the incubation of the agar plates was performed in an anaerobic atmosphere composed of 20% CO_2_ and 80% N_2_, at 37 °C for 48 h in a HEPA CLASS 100 Thermo Electron incubator (Thermo Electron Corporation, Marietta, OH, USA). 

### 2.4. Analytical Methods

#### 2.4.1. Standard Enological Parameters 

Ethanol, pH, titratable acidity, volatile acidity and reducing substances)were measured in the wines, according to the official methods established by the International Organization of Vine and Wine [27]. 

#### 2.4.2. The Total Polyphenols Content

The total polyphenols content was measured by the modified Folin–Ciocalteau method described originally by Singleton and Rossi in 1965 [28], after our own modifications. In brief, 0.3 mL of the tested wine samples (after proper dilution) were mixed with 4.15 mL deionized water, 0.05 mL of Folin-Ciocalteu reagent (Sigma-Aldrich, Munich, Germany), and 0.5 mL of 5% 2 M sodium decarbonate solution (POCh, Gliwice, Poland) and mixed. After 20 minutes of incubation at room temperature, the absorbance of the samples was determined at 700 nm. The data were calculated as gallic acid equivalent per liter of wine (g GAE/L; Sigma-Aldrich, Munich, Germany). 

#### 2.4.3. The Antioxidative Activity 

The antioxidative activity was determined against the ABTS reagent, according to the method described by Re et al. 1999 [29]. In briefl 3 mL of ABTS reagent (Sigma-Aldrich, Munich, Germany) and 30 µL of sample were measured into 10 mL test tubes. The tubes were tightly closed, mixed thoroughly, and set aside in the dark for 6 min at 21 °C. After the incubation, spectrophotometric analysis (Halo SB-10 Dynamica Biogenet, Cambridge, UK) was performed at 735 nm. Results of the assay were expressed as the capability of antioxidants to scavenge ABTS radicals, relative to that of Trolox (Sigma-Aldrich, Munich, Germany), and were presented as mM of Trolox equivalent per liter of wine (mM TE/L).

#### 2.4.4. Alpha Glucosidase Inhibition Assay

Alpha glucosidase inhibition assay was adapted from Czapska-Pietrzak et al., 2019 [30] and Johnson et al., 2011 [31]. The test is based on the measurement of the absorbance of the sample in which, as a results of alpha glucosidase activity, a release of glucose and p-nitrophenol from PNPG (4-Nitrophenyl-β-D-glucopyranoside) in the alkaline environment is observed. The released p-nitrophenol (PNG) has yellow color and shows maximum absorption at wavelength = 405 nm. The stronger the alpha-glucosidase inhibition caused by the test components (wine), the lower the amount of p-nitrophenol released from PNPG as a consequence of alpha-glucosidase action. The applied enzyme concentration was 1 U/mL. To provide the alkaline conditions, 0.1 M phosphate buffer pH 6.8 (POCh, Gliwice, Poland) was used. The samples were prepared as follows: test sample, 50 μL of wine, 50 μL of buffer and 30 μL of alpha-glucosidase (from *Saccharomyces*, Sigma-Aldrich, Munich, Germany), and background to the test sample, 50 μL of wine and 80 μL of buffer. 

For the control sample, 50 μL of water, 50 μL of buffer and 30 μL of alpha-glucosidase, and as background to the control sample, 5 μL of water and 80 μL of buffer. The samples were incubated at 37 °C for 15 min, then 20 μL of 4 PNPG (Sigma-Aldrich, Munich, Germany) was added, and the sample incubated for 20 min. After this time the reaction was stopped by adding 0.1 M sodium carbonate (POCh, Gliwice, Poland). Absorbance of the samples was performed, and the results were calculated according to the following formula: % of inhibition = (A_cs_ − A_ts_)/A_cs_ × 100%

A_cs_: absorbance of control sample–absorbance of background of control sample,

A_ts_: absorbance of test sample–absorbance of background of test sample.

The relation between the tested wine concentration (in the range 5–100 μL) and the alpha-glucosidase inhibition potential was determined. As a result, the obtained logarithmic curves were described with mathematical formulas which were then used for the IC_50_ values calculation.

### 2.5. Statistical Analysis

All wines variants (wine A, B, C) were made in triplicates, whereas analyses of all parameters were performed in at least four repetitions. All data are presented as mean values ± standard deviations. Tukey’s multiple range test was applied for the determination of the significant differences between the data (*p* ≤ 0.05). Statistica 13.3 (StatSoft, Tulsa, OK, USA) was used for this purpose. 

## 3. Results and Discussion

### 3.1. Changes of Vinification Parameters during Winemaking

The dynamics of alcoholic fermentation was not affected during the vinification process when green tea was added to the cherry must (Figure 2). In all three vinification variants, both the dynamics and the efficiency of the process were very similar. The alcohol content in the final wines ranged from 10.54 to 11.25%, and no significant differences were found (Table 2). The dynamics of sugar use in all variants was also comparable. All the wines obtained were classified as dry, but statistically significant differences were noted in the concentration of reducing sugars after 6 months of winemaking. The lowest concentration of 1.31 g/L was recorded for the control wine. In wines produced with the addition of green tea infusion, significantly higher residual sugars (6.93 and 5.86 g/L for wine A and B, respectively,) were present (Table 2). This may indicate some inhibition of yeast metabolic activity during fermentation, caused by an increased concentration of polyphenolic compounds, known for their antimicrobial properties [32,33,34]. 

The three variants of wine must had different initial polyphenols concentrations, depending on the amount of tea used in the infusion applied. Similarly, significant differences in the initial antioxidant activity were noted. Both parameters were reduced during the winemaking process. This could be caused by the bioconversion of polyphenolic compounds to less active forms [35,36]. The reason could also be the adsorption of polyphenols to yeast cell walls or their binding with yeast cell proteins. A similar phenomenon was described [15,37] during the production of cherry wine. In those research studies, a 20 and 25%, respectively, reduction in the content of polyphenolic compounds during the vinification process was noted. In our experiment, the losses of polyphenolic compounds were 21–25%. Finally, the content of polyphenolic compounds in the control wine was 1.89 g GAE/L of wine. These values are comparable to those described in the literature [14], or higher [13,15,38]. It was noted, that wine variants with the addition of green tea contained significantly more polyphenols: 2.22 and 2.73 g GAE/L in wine A and B, respectively, (Table 2).

During the vinification process, a visible reduction in acidity was also noted. In fruit wines, this phenomenon most often occurs when malic acid (which is the dominant acid in fruit) is metabolized by the yeast used in the process. Such specific yeasts, aimed at reducing malic acid during winemaking in the presence of glucose and other assimilable carbon sources, have the ability to significantly reduce acidity (up to 90% of malic acid consumption). Wine yeasts *Saccharomyces cerevisiae* belong to this group, together with *Schizosaccharomyces pombe* and *Zygosaccharomyces bailii* [39]. In grape wines where tartaric acid is the dominant organic acid, the conditions of winemaking (e.g., cold stabilization, maturation) are an additional factor contributing to the reduction of acidity. Increasing the concentration of ethanol and gradually lowering the fermentation temperature, followed by maturation and cold stabilization, decreases the solubility of tartaric acid, causing its precipitation as potassium bitartrate. The wine yeast does not have the ability to degrade tartaric acid in its biochemical pathway [40,41]. In this study, the reduction of acidity was only 0.41–0.45 g/L, which was significantly lower than that described by Sun et al. 2011 [15]. This may indicate a lower malic acid utilization activity of yeast used in this experiment. 

### 3.2. Changes in Total Polyphenolics Content during In Vitro Gastrointestinal Digestion

The produced experimental wines were subjected to the in vitro digestion process. Changes in polyphenols concentration and antioxidant activity during the successive stages of digestion process were analyzed. The initial concentration of polyphenols was 1.89 g GAE/L for the control wine, and then 2.22 and 2.73 g GAE/L for wines A and B, respectively, (Table 2). The first stage of digestion is the stomach zone. At this stage, incubation in pH 2.0 with pepsin for 2 h resulted in a visible increase in polyphenol concentration by 10, 13 and 16% for wines B, A and C, respectively, (Figure 3). In the following zone of digestion, the acidity of the stomach was gradually reduced from pH 2 to pH 6, and the food was digested in the presence of pancreatin and bile salt. This allows the pH to increase further, and such conditions (pH 7.4) imitate the small intestine. Samples taken at this point had lower concentrations of polyphenols compared with those in the stomach zone. After 18 h of incubation in the large intestine, another significant reduction in the concentration of polyphenols was noted. The decreasing trend observed in total phenolic content along the digestion chain was also observed for antioxidant activity (Figure 4).

In general, during the performed in vitro gastrointestinal digestion, the concentration of total polyphenols was reduced in the range of 53–64%, and reduction of antioxidant activity occurred in the range of 38–45%. The percent reduction of total phenolic concentration after wine in vitro digestion is in agreement with other research groups [42,43,44]. According to the published literatures, such high losses of polyphenols may be caused by drastic pH changes during the digestion process. The results of in vitro digestion of cherry wine [45], as well as other products [46,47], suggest that polyphenolic compounds are unstable under neutral and alkaline environments, as opposed to those of high acidity. This may explain the significant increase in their concentration in the stomach stage followed by the intense decrease in the small and large intestines. Donlao and Ogawa, 2018 [48] maintained that most green tea polyphenols are relatively stable in the gastrointestinal environment. However, they also noted in their studies a significant reduction in antioxidant activity in the range of 16–25%, evaluated as DPPH, which agrees with the 27% antioxidant activity reduction reported by Record and Lane 2001 [49]. Our findings indicate that antioxidant activity reductions are much more severe compared with the reported findings above, but in our case green tea infusion was just an addition to cherry wine, which makes the product of a different specification. Nevertheless, in all the above-mentioned studies, close relationships between the antioxidant activity, the concentration of polyphenols and the acidity of the environment were observed. For example, when analyzing selected groups of polyphenols, a significant increase in flavanol concentration after stomach digestion was noticed. This was a result of proanthocyanidin hydrolysis in the low pH conditions [43].

A similar phenomenon was described by [47] when analyzing the bioaccesability of grape polyphenols. Additionally, they showed that 88.8% of all polyphenols extracted after the 2 h gastric digestion stage were flavonoids, and the other 11.2% compounds were mainly phenolic acids. In the next step, in the mild alkaline intestinal environment, similarly to our results, a significant reduction of bio-accessible phenolic acids, flavonoids and anthocyanins was observed. 

Another important aspect is the influence of digestive enzymes that release polyphenols from the food matrix, making them more susceptible to hydrolysis [47,50,51]. Moreover, the fluctuation of polyphenol content may also be the result of enzymatic activity of the intestinal microflora. The beta-glucosidases, beta-glucuronidases, rhamnosidases and esterases produced by these, cause the degradation of digestible phenolic compounds, thus influencing their qualitative and quantitative profile. In our previous study [25] a significant reduction in the concentration of phenolic compounds in grape and chokeberry wines was also observed at the last stage of the digestion process. Therefore, we suggested that the reason for this phenomenon was the activity of digestive enzymes, which are responsible for the hydrolysis of glycosidic bonds followed by the formation of aglycone forms and then the derivatives generation. As a result, the hydrolysis products are more susceptible to biotransformation by the intestinal microflora, and can be used by bacteria in their metabolic processes. This all has a significant impact on the intestinal environment, modulating the gut microbiota system and, consequently, their functional effect in benefits to the host [5,52].

### 3.3. Changes in Intestinal Microflora during In Vitro Digestion

During digestion of the tested wines, the growth of the intestinal microflora was also monitored. Four test strains (*Lactobacillus plantarum*, *Enterococcus faecium*, *E. coli* and *Bifidobacterium bifidum*) were selected for the study as representatives of gut microbiota from the family of *Lactobacilliaceae*, *Enterobacteriaceae*, *Bifidobacteriaceae* and *Enterococcaceae*, respectively. However, the human intestine is a dynamic microbial ecosystem inhabited by a highly concentrated (up to 10^12^ cells/gram of faces) mixed bacteria culture. Its qualitative and quantitative profile depends on many factors, such as age, diet, lifestyle, diseases, and others. Therefore, it is relatively difficult to unequivocally assess the influence of various factors on the growth of intestinal bacteria; however, such studies are very much needed. In our experiment, in the case of the control wine, no inhibition of, and even a slight increase in, the cell count of *Lactobacillus* and *Bifidobacterium* was noted. On the other hand, the growth of *E. faeciun* and *E. coli* were significantly inhibited, while *E. coli* was definitely more so—approximately 2.5 logarithmic units (Figure 5.) In other studies, *Lactobacillus* bacteria assessed during the digestion of red wine [53] also showed the ability to grow in the presence of wine polyphenols. The authors suggest that phenolic metabolites formed as a result of digestive processes may contribute to the stimulation of *Lactobacillus*. Additionally, they draw attention to the fact that some *lactobacilli*, such as *L. acidophilius* and *L. plantarum*, have the ability to improve the production of butanoic acid, whose beneficial effect on the intestinal wall and the entire bacterial flora is widely recognized, especially in the field of reduction of oxidative stress, inflammation, and its anticarcinogenic activity [54]. 

We noticed that *E. coli* were inhibited with a similar intensity during the digestion of the control wine and wine A (with a lower concentration of tea infusion), where the average final reduction in the number of bacteria was approximately 2 log units (Figure 5). It can be assumed that polyphenolic compounds derived from tea at this concentration did not determine the growth of bacteria. We presume that the inhibitory factor in this case could be, e.g., ethanol. Similar observations were reported by Cueva et al. 2015 [53], who noticed that *lactobacilli* were inhibited by ethanol from wine. However when there was a higher concentration (approximately three times higher) of tea infusion, a significantly higher concentration of polyphenols in wine B contributed to a radical reduction (5.07 log units) in the number of *E. coli*. The antibacterial ability of phenolic compounds is also explained by other researchers as an effect of bacterial proliferation inhibition, which in consequence causes the reduction of bacterial counts, not necessarily destroying the cells, but efficiently preventing the colony or biofilm formation [8,9]. In general, a significantly greater inhibition of the growth of intestinal bacteria was noted during the digestion of wines with concentrated green tea. Wine B, prepared with concentrated green tea infusion, showed the strongest inhibiting properties. The growth of all tested groups of bacteria was the most limited in this wine. *E. coli* were the most sensitive microorganisms. Their amount was reduced from the initial 6.16 log cfu/mL to 1.09 log cfu/mL after digestion of wine B was completed (Figure 5). 

A study on the influence of grape and chokeberry wine on intestinal microflora indicated that bacteria of the *Enterobacteriaceae* and *Enterococcus* families showed high sensitivity to wine polyphenols [25]. Another research study on the profile of changes in selected polyphenolic compounds allowed the analysis of the interaction of microflora-polyphenols [52]. Almost complete degradation of chlorogenic acid was explained by these authors as the ability of *E. coli*, *Lactobacillus* and *Bifidobacterium* to synthetize esterase, causing the degradation of chlorogenic acid followed by consumption of the obtained final metabolites by the tested bacteria [52]. Such interdependencies between individual polyphenolic compounds and the intestinal microflora are very helpful in understanding the functioning of the gastrointestinal tract and modulating the gut microbiota composition. Taguri et al. 2006 [55] studied the relationship between the structure of 22 polyphenols and their antibacterial activity against 26 bacterial species. They found a positive relationship between polyphenol structure (the number of pyrogallol rings) and antibacterial activity. Gallic acid, so important in our study as the calculating equivalent of total phenolic compounds in cherry wines, has one of these rings in its structure, and was classified as a compound with a moderate antimicrobial potential. Additionally, an involvement of gallic acid in the prevention of bacteria biofilm formation was presented. A high reduction (more than 70%) for the tested biofilms formed by *E. coli*, *P. aeruginosa*, *S. aureus*, *L. monocytogenes* and *S. mutants* were noted as results of gallic acid biological activity [56,57]. 

### 3.4. Inhibition of Alpha-Glucosidase Activity 

Polyphenolic compounds, as well products rich in them, have the ability to inhibit alpha-glucosidase activity. As a result, their presence in the diet can help control postprandial hyperglycemia by restricting glucose release and its adsorption into the blood. This allows the control of the insulin response and supports therapy for diabetes. Additionally, these inhibitors which come from natural sources such as plant-based food most often show even greater bioactivity compared with their synthetic analogues such as acarbose, miglitol or voglibose, while showing significantly less oppressive gastrointestinal side effects [21,58,59]. 

The wines produced in this study (with and without green tea infusion) were assessed for their potential to inhibit alpha-glucosidase. The influence of the in vitro digestion process on these wines, based on enzyme inhibition activity, was analyzed. To present the inhibitory effect, logarithmic trend curves were plotted and the IC_50_ values were calculated (Figure 6). The wines were tested in the concentration range 5–100 μL before and after in vitro digestion, to evaluate the influence of the digestion process on the AGL inhibitory activity. The trend of the curves was very similar for all wine types (Figure 6), but the IC_50_ values differed very clearly (Figure 7). 

When analyzing the amount of wine that would allow a 50% reduction in enzyme activity, the highest volume—meaning the lowest inhibitory activity, was recorded for the control wine—20.53 μL. Meanwhile, the highest activity was recorded for wine B (with the highest tea addition), where only 4.85 μL was enough to achieve the same effect (IC_50_). Wine A also showed a significantly higher inhibitory activity, compared with the control wine, IC_50_ = 8.06 µL. The in vitro digestion process modified the activity of wines towards the inhibition of AGL. For wine A, more than twice the amount of wine was needed to achieve half the reduction in enzyme activity, while for wine B and the control wine, the differences were smaller but statistically significant (Figure 7).

Since polyphenols are mainly indicated as the factor determining the activity of AGL inhibition [21,58,59] we carried out some calculations to determine the concentration of polyphenols that will result in a 50% reduction in AGL activity (Figure 8). The result was that in wine B, a 50% reduction was obtained using 13.24 μg/mL of polyphenols. Interestingly, after digestion of this wine, the concentration of polyphenolic compounds necessary to obtain the IC_50_ value was significantly lower, i.e., 8.34 μg/mL of polyphenols. The same relations were noted in the other variants of wines, with the difference in total polyphenol content in the control wine being clearly the greatest (38.8 μg/mL in wine before, and 15.13 μg/mL in wine after, digestion). 

Analyzing the influence of the digestion process on the ability to inhibit the action of AGL, it was shown that to obtain a 50% reduction in the enzyme activity, more wine (volumetrically, μL) was needed after digestion than before (Figure 7). However, by calculating the content of polyphenols in these volumes of wines, it was determined that fewer polyphenols (quantitatively, μg) were needed after digestion (Figure 8). The phenomenon most likely results from the specificity of the bioconversion of polyphenolic compounds which, as a result of digestion, were transformed into compounds with significantly higher biological activity (towards AGL inhibition). From a nutritional point of view, these are extremely promising observations. Despite a very significant reduction in the content of polyphenols recorded at subsequent stages of in vitro digestion (Figure 3), it has been shown that their biological activity towards AGL inhibition is even greater—fewer polyphenols are needed to obtain a 50% reduction in enzyme activity (Figure 8). In order to explain this phenomenon, investigations by other researchers who tested the inhibition activity of AGL using polyphenolic compounds in different models, were compared. Cakar et al. 2019 [18] evaluated the AGL inhibitory effect induced by phenolic compounds from cherry, raspberry, blackberry and blueberry wines. The highest anti-enzyme activity was noted for ellagic acid, chlorogenic acid and catechin. In general, from the group of hydroxybezoic acid derivatives, protocatechuic and gallic acids also showed high inhibiting potential, which is in agreement with Oboh et al., 2015 [60]. In another study comparing different phenolic acids, it was found that caffeic acid presented higher inhibition activity than acarbose, while the lowest inhibition was shown by syringic acid (a cinnamic acid derivative) and vanillic acids (benzoic acid derivative) [20,61]. Analyzing the chemical structure of these compounds, the authors suggest that hydroxyl groups might be crucial in the inhibition effect. Compounds containing in their structure more than one hydroxyl group (caffeic acid, protocatechuic acids) were characterized with higher inhibition activity in comparison to those with one or no hydroxyl group, as well as compounds with methoxy groups (p-coumaric, syringic and vanillic acids) [6,60]. Moreover, polyphenolic compounds differ in polarity, which also affects their potential to inhibit the enzyme. It is suggested that, to block the activity of alpha-glucosidase, a specific reaction between two phenolic compounds is needed. To reach the enzyme blocking effect a dipol-dipol interaction must occur, and in addition, certain conditions must be met—the dipolar moment must be high, and the dipole must be formed close to the enzyme active site [20]. Polyphenols can interact with each other and enhance their biological activity in synergistic interactions, but they can also react with other food ingredients. Such binding with the food matrix as, for example, that of a phenolic-protein molecule, can exhibit decreased biological activity [62]. However, under the conditions of simulated gastrointestinal digestion, it was demonstrated that phenolics were gradually released from complex molecules, and their biological activity (antioxidant as well as enzyme inhibition activity) increased accordingly [63]. 

It is noteworthy, however, that the exact comparison of AGL inhibitory properties is not entirely possible. The influence of enzyme type (from yeast or rat small intestine), polysaccharide, plant extract substrate and its preparation, as well as in vitro conditions and reagents used, are essential for the obtained results. Aleixandre et al. 2022 [20] found that different ways of substrates pretreatment significantly determined the final results of AGL inhibition capacity. In their experiment, phenolic acids were previously incubated with the enzyme, as well as with the starch, before and after gelatinization. Significant differences (although higher for alpha-amylase then for alpha-glucosidase inhibition) were described. On the other hand, it was showed that the type of alpha-glucosidase enzyme used in the test analysis is very important. The same phenolic compounds—proanthocyanidins, showed strong inhibition against alpha-glucosidase activity when an enzyme isolated from yeast was applied [64,65], and no inhibition effect when rat intestinal alpha-glucosidase was applied [66]. Similarly, higher AGL inhibition for aglycones of flavonoids [67] and pyrolle alkaloid extracts [68] were noted when an enzyme derived from *Saccharomyces cerevisiae* was used.

## 4. Conclusions

To our knowledge, this is the first report on this type of fruit wine-cherry wine enriched with green tea bioactive compounds. Our results indicate possibilities in developing this new product with enhanced pro-healthy properties. 

The addition of green tea to the cherry wine significantly increases the total polyphenol content and antioxidant activity, and concurrently does not cause any disturbances in the course and dynamics of the vinification process. During in vitro digestion of the produced wines, interesting observations have been noted. Firstly, a significant increase in polyphenolic concentration in the stomach stage, followed by an intense decrease in the small and large intestines zone. This can suggest that polyphenolic compounds are unstable within a neutral and alkaline environment, as opposed to those of high acidity. 

In the human intestine, a dynamic microbial ecosystem reacts differently to the digested food compounds, especially those with antimicrobial properties (such as polyphenols). Interaction between the evaluated cherry wines and intestinal microflora presented both inhibition and stimulation of the microorganisms growth. From our observations, *E. coli* was the most sensitive bacteria, while the most resistant were *L. plantarum* and *B. bifidum*. 

The tea-derived bioactive compounds also significantly influenced the increase of the potential of the AGL inhibition in the wines. Moreover, an even higher AGL inhibition potential was observed in wines after the digestion process, which, from a nutritional point of view, is a very promising aspect. 

In conclusion, the proposed type of cherry wine could be a good alternative beverage with the added value of a high polyphenolic content, high antioxidant activity, and very high activity for alpha-glucosidase inhibition, making it a natural diet component with increased potential for diabetes prevention.

## Figures and Tables

**Figure 1 foods-11-03298-f001:**
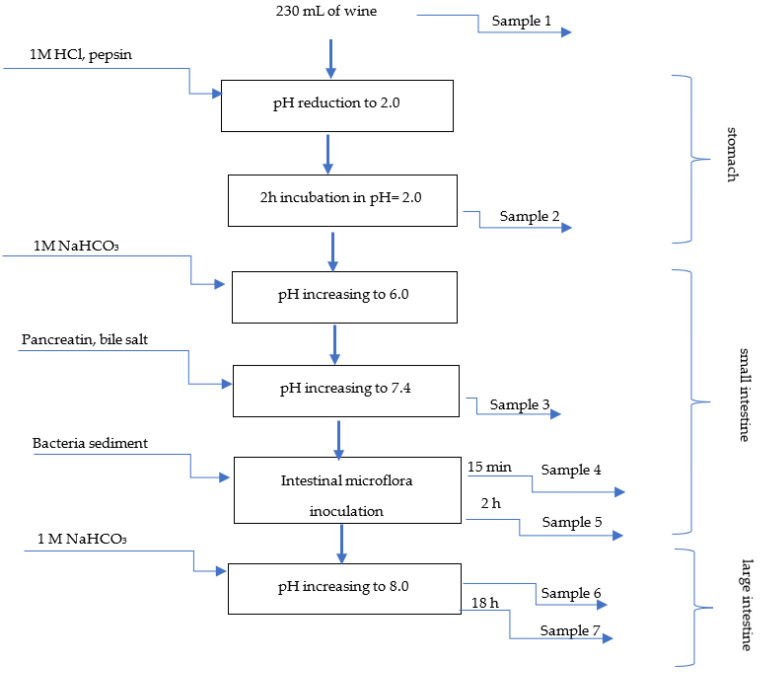
Diagram of in vitro gastrointestinal digestion model and sampling stages.

**Figure 2 foods-11-03298-f002:**
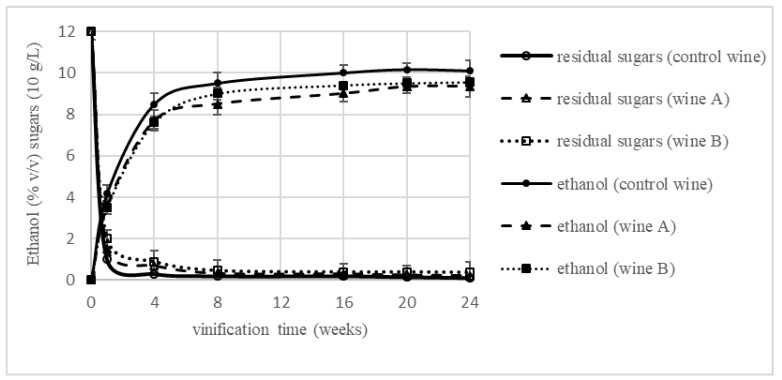
Sugar consumption and ethanol production during vinification of cherry wine with green tea addition.

**Figure 3 foods-11-03298-f003:**
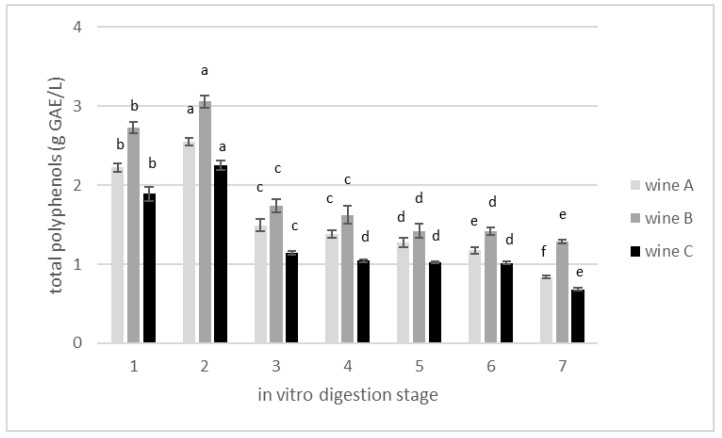
Changes in total polyphenol concentration expressed as gallic acid equivalent (g GAE/L) in cherry wines during in vitro digestion. 1—wine before digestion; 2—after stomach; 3—small intestine, without microflora; 4—small intestine, 15 min after intestinal microflora inoculation; 5—small intestine, 2 h after intestinal microflora inoculation; 6—large intestine after reaching pH 8.0; 7—after large intestine, end of digestion. Different letters (a,b,c,d,e,f) mean significant differences (*p* ≤ 0.05).

**Figure 4 foods-11-03298-f004:**
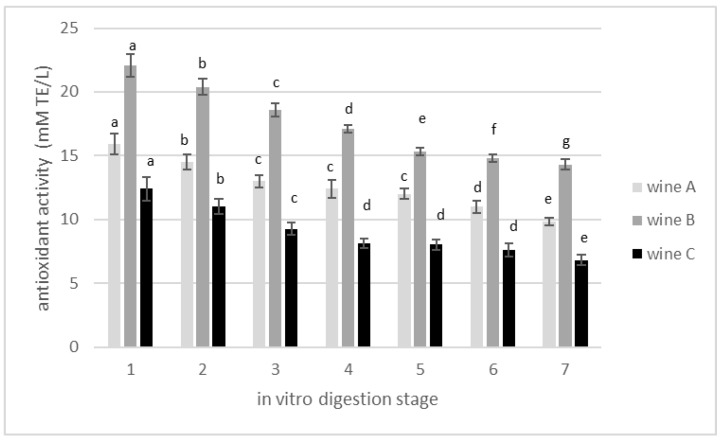
Changes in antioxidant activity expressed as mM Trolox equivalent (mM TE/L) in cherry wines during in vitro digestion. 1—wine before digestion; 2—after stomach; 3—small intestine, without microflora; 4—small intestine, 15 min after intestinal microflora inoculation; 5—small intestine, 2 h after intestinal microflora inoculation; 6—large intestine after reaching pH 8.0; 7—after large intestine, end of digestion. Different letters (a,b,c,d,e,f,g) mean significant differences (*p* ≤ 0.05).

**Figure 5 foods-11-03298-f005:**
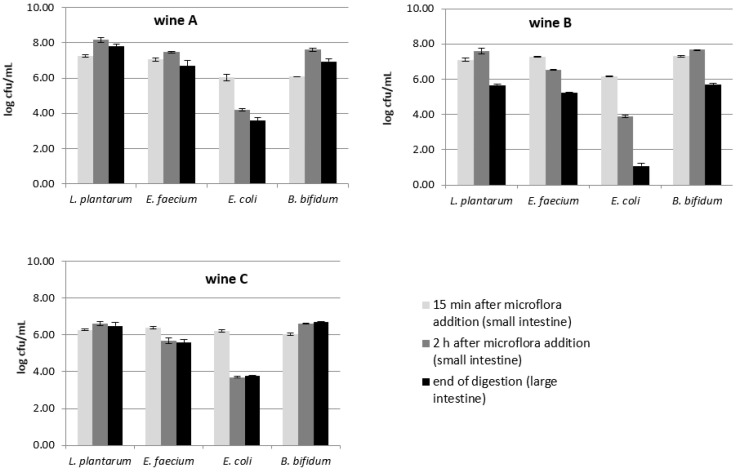
Changes in intestinal microflora concentration during in vitro digestion of the tested cherry wine variants.

**Figure 6 foods-11-03298-f006:**
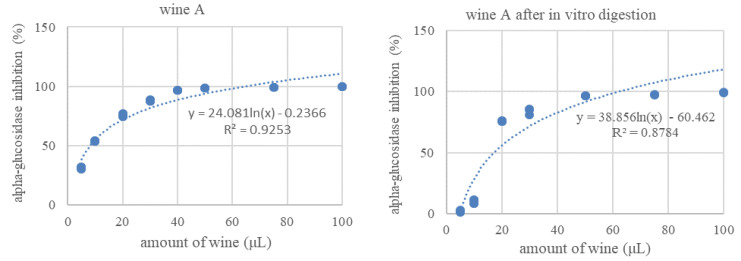
Inhibitory effect of the tested cherry wines against alpha-glucosidase activity before and after in vitro digestion.

**Figure 7 foods-11-03298-f007:**
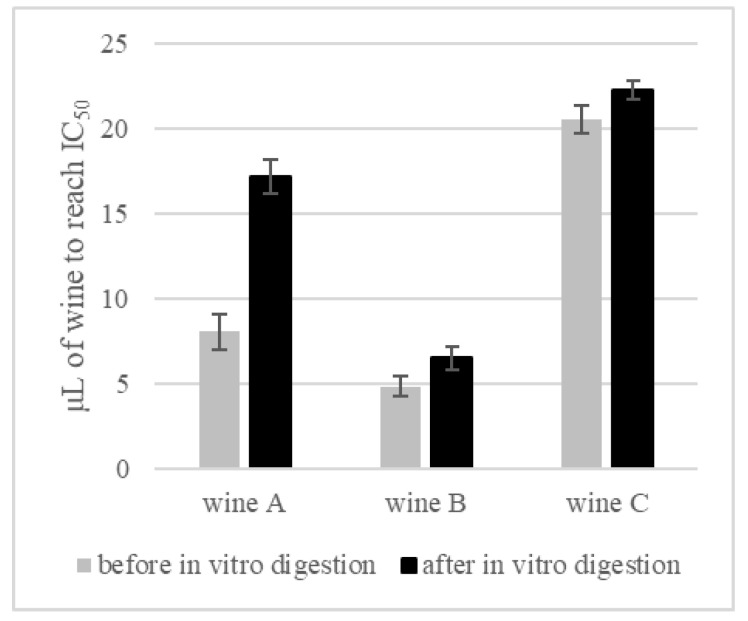
Amount of wine (μL) needed to obtain 50% inhibition of alpha-glucosidase (IC_50_).

**Figure 8 foods-11-03298-f008:**
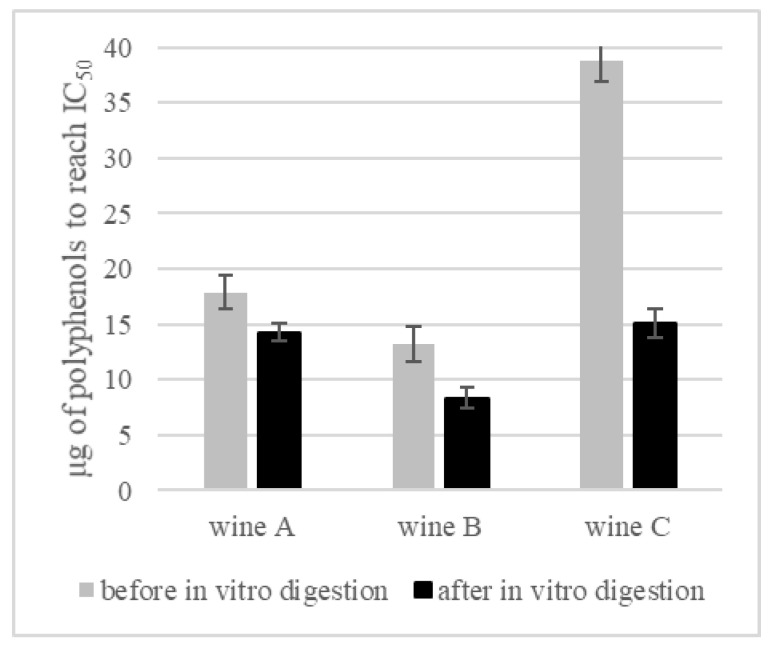
Amount of polyphenols (μg) needed to obtain 50% inhibition of alpha-glucosidase (IC_50_).

**Table 1 foods-11-03298-t001:** Composition of cherry must.

Ingredient	Variant of Wine		
	Wine A (“with Green Tea Infusion 7 g/L”)	Wine B (“with Concentrated Green Tea Infusion 21 g/L”)	Wine C (“Control Wine”)
Cherries	12 ± 0.1 kg	12 ± 0.1 kg	12 ± 0.1 kg
Sugar (sucrose)	1.4 ± 0.01 kg (beginning) 1.4 ± 0.01 kg (after 1 week)	1.4 ± 0.01 kg (beginning) 1.4 ± 0.01 kg (after 1 week)	1.4 ± 0.01 kg (beginning) 1.4 ± 0.01 kg (after 1 week)
Water	not applied	not applied	10 ± 0.1 L
Green tea infusion	10 ± 0.1 L *	10 ± 0.1 L **	not applied
K_2_S_2_O_5_	50 ± 0.01 mg/L	50 ± 0.01 mg/L	50 ± 0.01 mg/L
nutrition	0.25 ± 0.01 g/L	0.25 ± 0.01 g/L	0.25 ± 0.01 g/L

* 70 g of dry green tea leaves were used for 10 L of water, three-times infusion was performed (final concentration in the tea infusion = 7 g/L), ** 210 g of dry green tea leaves were used for 10 L of water, three-times infusion was performed (final concentration in the tea infusion = 21 g/L).

**Table 2 foods-11-03298-t002:** Chemical characteristic of musts and cherry wines.

Parameters	Wine A		Wine B		Wine C	
	Must	Wine	Must	Wine	Must	Wine
Ethanol (% *v*/*v*)	n.d.	10.97 ± 0.66 ^a^	n.d.	10.54 ± 0.71 ^a^	n.d.	11.25 ± 0.83 ^a^
Reducing sugars (g/L)	121.64 ± 3.34 ^a^	6.93 ± 0.33 ^c^	123.48 ± 2.96 ^a^	5.86 ± 0.54 ^b^	121.26 ± 1.68 ^a^	1.31 ± 0.29 ^a^
pH	3.69 ± 0.06 ^a^	3.77 ± 0.08 ^a^	3.71 ± 0.08 ^a^	3.79 ± 0.09 ^a^	3.64 ± 0.04 ^a^	3.74 ± 0.09 ^a^
Titratable acidity (g/L) *	8.24 ± 0.11 ^b^	7.79 ± 0.08 ^b^	8.39 ± 0.11 ^b^	7.95 ± 0.11 ^b^	8.09 ± 0.12 ^a^	7.68 ± 0.07 ^a^
Volatile acidity (g/L) **	n.d.	0.26 ± 0.01 ^b^	n.d.	0.17 ± 0.01 ^a^	n.d.	0.31 ± 0.02 ^c^
Total polyphenols (g GEA/L) ***	2.97 ± 0.08 ^b^	2.22 ± 0.03 ^b^	3.47 ± 0.12 ^c^	2.73 ± 0.07 ^c^	2.43 ± 0.04 ^a^	1.89 ± 0.02 ^a^
Antioxidant activity (mM TE/L) ****	17.04 ± 0.65 ^b^	15.92 ± 0.51 ^b^	23.71 ± 0.44 ^c^	22.07 ± 0.66 ^c^	14.78 ± 0.61 ^a^	12.39 ± 0.47 ^a^

Wine A: cherry wine with green tea infusion addition (7 g/L), Wine B: cherry wine with concentrated green tea infusion addition (21 g/L), Wine C: control cherry wine, n.d.—not detected, * as malic acid; ** as acetic acid; *** as gallic acid equivalent; **** as Trolox equivalent, Different letters in the rows (a, b, c) mean significant differences (*p* ≤ 0.05), evaluated separately for wine samples and must samples.

## Data Availability

The data presented in this study are available on request from the corresponding author.

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
