# Peer review of "Influence of Green Tea Added to Cherry Wine on Phenolic Content, Antioxidant Activity and Alpha-Glucosidase Inhibition during an In Vitro Gastrointestinal Digestion"

_foods, 2022, doi:10.3390/foods11203298_

Round 1
Reviewer 1 Report
1- The title could be modified to :Influence of addation green tea on the , antioxidant activity and alpha-glucosidase inhibition dur- 3 ing an in vitro gastrointestinal digestion of Cherry wine istead of Influence of green tea added to cherry wine on phenolic con- 2 tent, antioxidant activity and alpha-glucosidase inhibition dur- 3 ing an in vitro gastrointestinal digestion
2- The aime of the work line 83-88 is not clear should be rewritten
3- table one the standrad diviation should be added
4- page 6 line 195 Numbering must be added in all sub title such as 2.4.1 infront total phenolic----, 2.4.2 The antioxidative activity ----- , 2.4.3 Alpha glucosidase inhibition assay -----
5- in table 2 what is the meaning of small a,b,c ,d inside the table
6- . Conclusions need to improve
Author Response
Dear Reviewer,
Thank you for your accurate comments that allow us to improve the text and prepare for publication. We responded to all comments and suggestions. Below we present (in blue) point-by-point corrections, comments and responses for the Reviewer.
Comments and Suggestions for Authors
1- The title could be modified to :Influence of addation green tea on the , antioxidant activity and alpha-glucosidase inhibition during an in vitro gastrointestinal digestion of Cherry wine istead of Influence of green tea added to cherry wine on phenolic content, antioxidant activity and alpha-glucosidase inhibition during an in vitro gastrointestinal digestion
Answer: we decided not to change the title of the manuscript, after re-analyzing the suggestion of the Reviewer, as authors we stay with the opinion that the title in this form presents the content of the manuscript clear. So we would like to stay with the original title, and would like to ask You as Reviewer to accept it.
2- The aime of the work line 83-88 is not clear should be rewritten
Answer: this fragment was rewritten – line 75-79
3- table one the standrad diviation should be added
Answer: values of standard deviations were added to Table 1
4- page 6 line 195 Numbering must be added in all sub title such as 2.4.1 infront total phenolic----, 2.4.2 The antioxidative activity ----- , 2.4.3 Alpha glucosidase inhibition assay -----
Answer: All subtitles have been introduced (in “Materials and methods” as well as “Results and discussion”)
5- in table 2 what is the meaning of small a,b,c ,d inside the table
Answer: Different letters in the rows (a, b, c) mean significant differences (p≤0.05), line 230
6- . Conclusions need to improve
Answer: the whole conclusion chapter has been rewritten to better highlight the results of the study, line 454-472.
We hope that the manuscript meets now the journal's desired standards, and that after the introduced corrections you will find it suitable for publication.
Yours sincerely,
Małgorzata Lasik-Kurdyś, PhD,
Poznan University of Life Sciences,
Faculty of Food Science and Nutrition, Department of Fermentation and Biosynthesis
tel. +4861 848 72 88; fax. +4861 848 73 14; email: malgorzata.lasik@up.poznan.pl

Reviewer 2 Report
Lasik-Kurdys et al. have aimed to assess the biological properties of cherry wine made with the addition of green tea infusion. Several vinification parameters and biological activities such as antioxidant activity and alpha-glucosidase inhibition potential were investigated and presented well in a systematic way. In general this article is written well and the results obtained are appropriately interpreted and conclusions justify the hypothesis tested. However the following points need to be addressed before this paper could be accepted for publications in Foods: 1. Include the quantitative data for inhibition effect on intestinal microflora and alpha-glucosidase inhibition effect. 2. In the section 2, all the chemicals/instruments purchase should be accompanied by the state, city and country details in parenthesis in the case of USA, while city and country name for other countries. 3. Line 101, what is wit. B1? 4. Let the units usage be consistent. Like liters or L in text, tables and figures. Also microlitre usage with microsymbol with L for liter in text, tables.and figures. 5. Always separate the value and unit with a space. 6. Consider revising the title of section 2.3.3. 7. Provide the state/city & country details for Statistica 13.3. 8. Why significant difference is not provided for "must" in Table 2. 9. It is better to abbreviate the microbes names in x-axis of Fig.5 as LP, EF, EC, & BB and provide the full form in the figure caption. 10. "lactobacilli" or "Lactobacilli" with italics in line 386 and 401 and other places. 11. Be consistent with "alpha glucosidase" or "alpha-glucosidase". 12. A schematic diagram showing the take home points of results of parameters tested should be interesting before the conclusion.Author Response
Lasik-Kurdys et al. have aimed to assess the biological properties of cherry wine made with the addition of green tea infusion. Several vinification parameters and biological activities such as antioxidant activity and alpha-glucosidase inhibition potential were investigated and presented well in a systematic way. In general this article is written well and the results obtained are appropriately interpreted and conclusions justify the hypothesis tested. However the following points need to be addressed before this paper could be accepted for publications in Foods:
- Include the quantitative data for inhibition effect on intestinal microflora and alpha-glucosidase inhibition effect.
Answer: according to the inhibition effect on intestinal microflora – we present data of general changes in bacteria amount (Figure 5) because, in this way, the effect of stimulating the growth of these microorganisms in digestion conditions can also be presented as well as discussed in the manuscript.
The effect of alpha-glucosidase inhibition has been presented in Figure 6. Next, Figures 7 and 8 present as well the potential of inhibition alpha-glucosidase by the tested wines but from the point of view of the IC50 parameter. This allowed to compare quantitatively the amount of the tested wine (Figure 7) as well as the amount of polyphenolic compounds (Figure 8) which allow a 50 % reduction of the enzyme.
We decided to present the data in form of Figures, not in the Tables, to have possibility for better illustration of the profile of changes, which is much more difficult when data are collected in the table.
- In the section 2, all the chemicals/instruments purchase should be accompanied by the state, city and country details in parenthesis in the case of USA, while city and country name for other countries.
Answer: all the suggested information has been completed.
- Line 101, what is wit. B1?
Answer: it is thiamine, it was a component of nutrition media for yeast growth. To make it more clear some additional information have been introduced to the text (line 89-92).
- Let the units usage be consistent. Like liters or L in text, tables and figures. Also microlitre usage with microsymbol with L for liter in text, tables.and figures.
Answer: yes, all units have been entered correctly.
- Always separate the value and unit with a space.
Answer: yes, it was corrected.
- Consider revising the title of section 2.3.3.
Answer: We changed the subtitle from “Evaluation of bacterial cfu/mL” on “Determination of the number of bacteria” (line 161) – this title after corrections describes more accurately the essence of the analysis.
- Provide the state/city & country details for Statistica 13.3.
Answer: this information has been completed.
- Why significant difference is not provided for "must" in Table 2.
Answer: this data have been introduced, Table 2 has been completed.
- It is better to abbreviate the microbes names in x-axis of Fig.5 as LP, EF, EC, & BB and provide the full form in the figure caption.
Answer: yes, thank you very much for suggestion, it was corrected according directions.
- "lactobacilli" or "Lactobacilli" with italics in line 386 and 401 and other places.
Answer: yes, it was corrected in the pointed places as well as the whole manuscript was proofed according to this issue.
- Be consistent with "alpha glucosidase" or "alpha-glucosidase".
Answer: yes, it was corrected, sorry for this inattention.
- A schematic diagram showing the take home points of results of parameters tested should be
interesting before the conclusion.
Answer: yes, the whole conclusion chapter has been rewritten to better highlight the results of the study.
We hope that the manuscript meets now the journal's desired standards, and that after the introduced corrections you will find it suitable for publication.
Yours sincerely,
Małgorzata Lasik-Kurdyś, PhD,
Poznan University of Life Sciences,
Faculty of Food Science and Nutrition, Department of Fermentation and Biosynthesis
tel. +4861 848 72 88; fax. +4861 848 73 14; email: malgorzata.lasik@up.poznan.pl

Reviewer 3 Report
Recommendation: Major Revision
The manuscript influence of green tea added to cherry wine on phenolic content, antioxidant activity and ,alpha-glucosidase inhibition during an in vitro gastrointestinal digestion, the methodology was reasonable and technically sound. Here are some issues to be addressed.
Major concerns:
Comments to the Author:
The manuscript's title is appropriate. The main procedure and findings of the study are well expressed. Introduction: A brief survey of existing literature, the purpose, importance, and innovation of the research is well mentioned.
Point 1:Line 19 gGAE/l should be changed to g GAE/L. The article should be reviewed in its entirety according to the author's instructions in the journal Foods.
Point 2: At the beginning of the abstract, mention the importance of cherry wine and add more numerical sentences.
Point 3 line 100 Fermentation container information should be given.
Point 4: How were these amounts determined for green tea infusion?
Point 5: Plastic fermentation buckets should be the same as the description in the method before.
Point 6: Where and how were the samples stored to prevent variation in results?
Point 7: Line 128 Replace in vitro gastrointestinal digestion process - should be italicized
Point 8: Please review Figure 1 again. Explanation codes of gastrointestinal steps are not suitable. Lines 174-180 should be removed.
Point 9:Line 183 Add countries of the media
Point 10 In statistical analysis, I could not make sense of the least expression in repetition analysis. Why wasn't a clear repetition done?
Point 11 Statistica 13.3 company information should be written.
Point 12 Was Tukey used in the ANOVA test?
Point 13 Error bars are not visible in Figure 2.
Point 14 Shouldn't the n.a expression in Table 2 be n.d?
Point 15 I suggest lettering Figure 3 to help us understand the statistical differences.
Point 16 While treatment expressions are used in the paintings, for example, treatment A is used in the figure, and wine A is used. Use the names of the examples in the same way. While treatment expressions are used in the paintings, for example, treatment A is used in the figure, and wine A is used. Use the names of the examples in the same way. Please review all tables and figures again.
Point 17 Figure 5 uses the abbreviated names of microorganisms.
Point 18 I think figure 6 may not be used.
Point 19 The conclusion part is weak and should be expanded. I recommend that they highlight the results of the study.
Author Response
Dear Reviewer,
Thank you for your accurate comments that allow us to improve the text and prepare for publication. We responded to all comments and suggestions. Below we present (in blue) point-by-point corrections, comments and responses for the Reviewer.
Reviewer 3:
The manuscript influence of green tea added to cherry wine on phenolic content, antioxidant activity and ,alpha-glucosidase inhibition during an in vitro gastrointestinal digestion, the methodology was reasonable and technically sound. Here are some issues to be addressed.
The manuscript's title is appropriate. The main procedure and findings of the study are well expressed. Introduction: A brief survey of existing literature, the purpose, importance, and innovation of the research is well mentioned.
Point 1:Line 19 gGAE/l should be changed to g GAE/L. The article should be reviewed in its entirety according to the author's instructions in the journal Foods.
Answer: this was corrected and the whole manuscript has been reviewed according to instruction for authors in Foods.
Point 2: At the beginning of the abstract, mention the importance of cherry wine and add more numerical sentences.
Answer: we pointed in the first sentence of abstract the importance of cherries as a source/material for winemaking. More information about cherries and cherry wine is posted in the introduction text, line 47-59.
Point 3 line 100 Fermentation container information should be given.
Answer: this information has been completed, line 110.
Point 4: How were these amounts determined for green tea infusion?
Answer: the concentration of tea was developed in such a way that the infusion significantly increased the concentration of polyphenols in wine and at the same time it will not dominate the taste and aroma of the wine.
Point 5: Plastic fermentation buckets should be the same as the description in the method before.
Answer: this information has been completed, line 95-96 and 110.
Point 6: Where and how were the samples stored to prevent variation in results?
Answer: this information has been introduced into the text – line 116-117. “(After 6 months of winemaking process, the wine samples were stored at 7 ℃ and were directly subjected to appropriate analyses (without freezing).”
Point 7: Line 128 Replace in vitro gastrointestinal digestion process - should be italicized
Answer: this was corrected and the whole manuscript was reviewed according to this issue.
Point 8: Please review Figure 1 again. Explanation codes of gastrointestinal steps are not suitable. Lines 174-180 should be removed.
Answer: the sampling stages have been removed. We decided that in fact this description was redundant and repeated - the same information can be found in the diagram (Figure 1).
Point 9:Line 183 Add countries of the media
Answer: this information has been completed.
Point 10 In statistical analysis, I could not make sense of the least expression in repetition analysis. Why wasn't a clear repetition done?
Answer: All wines variants (wine A, B, C) were made in triplicates whereas analyses of all parameters were performed in at least four repetitions (Line 208-209)
Point 11 Statistica 13.3 company information should be written.
Answer: this information has been completed (line 206-209).
Point 12 Was Tukey used in the ANOVA test?
Answer: yes, this information has been completed (line 206-209).
Point 13 Error bars are not visible in Figure 2.
Answer: the Figure 2 was improved so the error bars are visible.
Point 14 Shouldn't the n.a expression in Table 2 be n.d?
Answer: yes it was corrected.
Point 15 I suggest lettering Figure 3 to help us understand the statistical differences.
Answer: yes, Figure 3 has been corrected according suggestions. Additionally Figure 4 has been completed as well.
Point 16 While treatment expressions are used in the paintings, for example, treatment A is used in the figure, and wine A is used. Use the names of the examples in the same way. While treatment expressions are used in the paintings, for example, treatment A is used in the figure, and wine A is used. Use the names of the examples in the same way. Please review all tables and figures again.
Answer: yes, it was corrected, only one term (wine A, wine B or wine C) is now using in the whole manuscript, the previous form may have been confusing.
Point 17 Figure 5 uses the abbreviated names of microorganisms.
Answer: yes it was corrected.
Point 18 I think figure 6 may not be used.
Answer: Figure 6 shows the profile of inhibition of alpha-glucosidase. It is not a directly or proportional dependence, as it might seem. Logarithmic curves present the strength and intensity of inhibition of AGL activity by the tested wines. From a scientific point of view, for the analysis of this effect, Figure 6, in our opinion, provides interesting information (like in some cases – for wine A and B – almost 100% of AGL inhibition was obtained for 40 μL of wine. After digestion of wine B, 20 μL already gives this effect. A lot of information can be read from this chart and the graphic image simplify the interpretation.
We would like to ask you to accept our standpoint.
Point 19 The conclusion part is weak and should be expanded. I recommend that they highlight the results of the study.
Answer: the whole conclusion chapter has been rewritten to better highlight the results of the study.
We hope that the manuscript meets now the journal's desired standards, and that after the introduced corrections you will find it suitable for publication.
Yours sincerely,
Małgorzata Lasik-Kurdyś, PhD,
Poznan University of Life Sciences,
Faculty of Food Science and Nutrition, Department of Fermentation and Biosynthesis
tel. +4861 848 72 88; fax. +4861 848 73 14; email: malgorzata.lasik@up.poznan.pl

Round 2
Reviewer 3 Report
The authors made the necessary revisions. I think it can be published after the journal is edited according to the author's instructions.